# Comparison of Whole Salivary Cortisol and Interleukin 1-Beta Levels in Light Cigarette-Smokers and Users of Electronic Nicotine Delivery Systems before and after Non-Surgical Periodontal Therapy

**DOI:** 10.3390/ijerph191811290

**Published:** 2022-09-08

**Authors:** Abdulkareem A. Alhumaidan, Khulud A. Al-Aali, Fahim Vohra, Fawad Javed, Tariq Abduljabbar

**Affiliations:** 1Preventive Dental Sciences Department, College of Dentistry, Imam Abdulrahman Bin Faisal University, Dammam 34212, Saudi Arabia; 2Department of Clinical Dental Sciences, College of Dentistry, Princess Nourah Bint Abdulrahman University, Riyadh 11564, Saudi Arabia; 3Prosthetic Dental Sciences Department, College of Dentistry, King Saud University, Riyadh 11545, Saudi Arabia; 4Department of Orthodontics and Dentofacial Orthopedics, Eastman Institute for Oral Health, University of Rochester, Rochester, NY 14620, USA

**Keywords:** cortisol, electronic nicotine delivery systems, interleukin, periodontal disease, smoking, unstimulated whole saliva

## Abstract

There are no studies that have compared whole salivary cortisol (CL) and interleukin 1-beta (IL-1β) levels in cigarette-smokers (CS) and electronic nicotine delivery systems (ENDS)-users before and after non-surgical periodontal therapy (NSPT). The aim was to compare whole salivary CL and IL-1β levels in light CS and ENDS users before and after non-surgical periodontal therapy (NSPT). Self-reported current CS, ENDS users, and non-smokers were included. A questionnaire was used to collect demographic data. All patients underwent NSPT. Periodontal parameters (probing depth (PD], gingival index (GI], clinical attachment loss (AL], plaque index (PI], and marginal bone loss (MBL]) and whole salivary CL and IL-1β were measured at baseline. At 3-months of follow-up, clinical parameters and whole salivary CL and IL-1β were re-assessed. *p*-values < 1% were arbitrated as statistically significant. Fifty-four individuals (18 CS, 18 ENDS users, and 18 non-smokers) were included. Clinical AL, MT, PD, PI, and MBL were similar in all groups at baseline. At 12-weeks of follow-up, PI (*p* < 0.01) and PD (*p* < 0.01) were high in CS and ENDS-users than non-smokers. Among non-smokers, there was a statistically significant correlation between whole salivary cortisol and IL-1β levels at 12-weeks’ follow-up (*p* < 0.001). There was no difference in whole salivary cortisol and IL-1β levels in CS and ENDS users at baseline and at 12-weeks follow-up. At 12-weeks of follow-up, there was a significant reduction in IL-1β (*p* < 0.01) and CL (*p* < 0.01) than baseline. In light CS and ENDS users without periodontal disease, clinical periodontal parameters and whole-salivary CL and Il-1β levels remain unchanged after NSPT.

## 1. Introduction

The use of electronic nicotine delivery systems (ENDS) such as electronic cigarettes is relatively common among individuals who have either quit or are attempting to quit conventional combustible tobacco smoking [1]. There is also a general misapprehension that vaping is less perilous to systemic and oral health than cigarette smoking [1,2]. However, scientific evidence has proved otherwise. Studies [3,4,5] have shown that the etiopathogenesis of respiratory and cardiovascular diseases is directly linked with the use of ENDS. Moreover, from an oral-health point of view, studies [6,7,8] have shown that vaping jeopardizes the morphology and function of human gingival fibroblasts; and even flavorings in nicotine-free electronic liquids (e-liquids) are toxic to cells [8,9]. Non-surgical periodontal treatment (NSPT) using hand instruments such as ultrasonic scalers and curettes is commonly performed for the treatment of periodontal inflammatory conditions such as gingivitis and periodontitis [2]. Smoking habits negatively impact clinical responses to surgical and NSPT [10,11]; however, with emphasis on vaping, ALHarthi et al. [2], reported that ENDS-users and non-smokers respond favorably to NSPT in terms of reduction in clinical periodontal inflammatory parameters (probing depth (PD], gingival index (GI), clinical attachment loss (CAL), plaque index (PI)) compared with CS. This reflects that a consensus is yet to be reached regarding the influence of NSPT on the treatment of periodontal diseases among individuals using ENDS.

Cortisol (stress hormone) is a glucocorticoid produced by the adrenal cortex [12]. During episodes of psychological anxiety/stress, cortisol is released into the bloodstream [12]. According to Cakmak et al. [13] and Dubar et al. [14] presence of anxiety and depression is not mandated for the expression of cortisol in unstimulated whole saliva (UWS). Higher cortisol levels (CL) have been reported in UWS samples collected from patients with compared to without temporomandibular disorders, periodontitis, and peri-implantitis [14,15,16]. In a recent study, Zhang et al. [17] compared whole salivary CL among smokers and non-smokers with and without periodontitis. The results showed that whole salivary CL was significantly higher in smokers with periodontitis compared with non-smokers with a healthy periodontal status. Saliva also expresses raised levels of inflammatory proteins such as interleukin 1beta (IL-β) in CS and ENDS-users with periodontal inflammation [18,19,20], and NSPT has been shown to reduce the salivary concentration of IL-1β in such patients [18,19]. There are no studies that have compared and/or correlated whole salivary CL and IL-1β levels in CS and ENDS users before and after NSPT. It is hypothesized that NSPT improves clinical periodontal status and reduces whole salivary CL and IL-1β levels in CS and ENDS users.

The aim of the present study was to compare periodontal status and whole salivary CL and IL-1β levels in light CS and ENDS users before and after NSPT.

## 2. Materials and Methods

### 2.1. Institutional Review Approval

Guidelines documented in the Helsinki Declaration as revised in 2013 for experiments on humans were followed. Withdrawal at any stage of the investigation was associated with no form of penalty. All individuals were provided written information about routine oral hygiene maintenance such as brushing techniques and were also informed about the detrimental effects of vaping and smoking on oral and systemic health. The current study was reviewed and approved by the Ethical research committee of the Centre for specialist dental practice and clinical research (UDRC-017/2021). Only participants who had read and signed a consent form were included.

### 2.2. Study Location

The present study was performed between April and October 2021 at the Dental unit of a tertiary healthcare located in Riyadh, Saudi Arabia. All patients were residents of Riyadh, ArRiyadh province, Saudi Arabia.

### 2.3. Criteria for Eligibility

Self-reported current CS (individuals with a smoking history of at least 5 years) [21], ENDS-users (individuals that were solely using ENDS for the past 12 months) [20], and non-smokers (individuals that had never used any type of tobacco/vaping product) [22] were enrolled. Dual smokers, patients with self-reported systemic diseases such as diabetes mellitus (DM), cardiopulmonary disorders, respiratory diseases, and cancer, pregnant and/or nursing females, and patients that had received surgical and/or NSPT within 12 weeks were excluded. Complete edentulism, third molars, and supernumerary/remaining root remnants were not assessed.

### 2.4. Questionnaire

Demographic details were collected using a questionnaire. Demographic data comprised of the following parameters: (a) determination of age (in years); (b) determination of gender (male or female or prefer not to say); history of vaping and CS (determination of pack-years); (c) family history of vaping/smoking habit; (d) history of psychological disorders such as anxiety and/or depression; (e) tooth-brushing (once/twice daily); and (f) flossing (at least once daily). The questionnaire was administered to all participants by the supervisor/principal investigator (TA).

### 2.5. Blinding

Only the study supervisor/principal investigator was aware of the smoking/non-smoking and vaping status of the participants. Examiners involved with the clinical, radiographic, laboratory-based investigation, and statistical analyses were blinded to the smoking/non-smoking and vaping status of the participants.

### 2.6. Periodontal Parameters

Clinical and radiographic investigations were performed by one trained and calibrated examiner (YA; *Kappa* score 0.85). The PI [23], GI [23,24], clinical AL [25], and PD [23] were measured at the mesiobuccal, distobuccal, midbuccal, mesiolingual, midlingual, and distolingual sites. A sterile graded probe was used to assess PD and clinical AL. The corresponding clinical AL and PD values were recorded in millimeters. Loss of interproximal bone or marginal bone loss (MBL) was measured in millimeters on digital bitewing radiographs taken using the long-cone paralleling technique [26,27]. Number of missing teeth (MT) was also recorded. All clinical examinations were performed at baseline and at 12 weeks of follow-up, whereas radiographic examinations were exclusively performed at baseline.

### 2.7. Collection of Whole Saliva and Assessment of Cortisol and IL-1β Levels

To collect UWS samples, patients were instructed to come during early morning hours (between 7 and 8 a.m.) in a fasting state. Patients were seated on a comfortable chair in a quiet room and were given a gauged measuring cylinder that was connected to a disposable plastic funnel. Patients were instructed to allow saliva to accumulate in the mouth for five continuous minutes and refrain from jaw movements and swallowing during this time. At the end of five minutes, patients were slightly open their mouth and allowed the saliva to drool into the plastic funnel. The unstimulated whole salivary flow rate was immediately recorded by a trained investigator (KAA; Kappa score 0.82). The collected saliva was then transferred into a plastic tube with a lid. The UWS samples were centrifuged at 1500 rpm for five minutes in a cold room. The supernatant was stored in sterile plastic tubes with a lid (Fisherbrand™ Premium Microcentrifuge Tubes, Waltham, MA, USA) at −70 °C. All samples were assessed for CL within 48 h. Commercial ELISA kits were used according to manufacturers’ instructions to assess CL (RayBio^®^, RayBiotech Life, Inc., Atlanta, GA, USA) and IL-1β levels (RayBio^®^, RayBiotech Life, Inc., Atlanta, GA, USA) in UWS supernatants. The samples were evaluated in duplicates in microtiter plates and read at 450 nm using a microplate-reader (StatFax 2100, Awareness Tech. Inc., Palm City, FL, USA). The detection range of salivary cortisol and IL-1β levels were 100–1,000,000 pg/mL and 0.3–100 pg/mL. The whole salivary CL was assessed by a trained evaluator (YA; *Kappa* score 0.84). The aforementioned protocols have been used in previous studies [20,28,29]. Collection of UWS and assessment of cortisol and IL-1β levels were performed at baseline and at 12 weeks of follow-up.

### 2.8. Non-Surgical Periodontal Treatment

The NSPT was performed 24 h after periodontal examination and UWS collection. An experienced investigator (FV) performed NSPT in all patients using an ultrasonic scaler (Dental Equipment Woodpecker Uds-J Ultrasonic Scaler EMS Compatible Original, Guangzhou, China) and sterile curettes (Hu-Frieddy, Chicago, IL, USA). All patients were instructed to rinse twice daily with 0.12% chlorhexidine gluconate (CHX) mouthwash and routine oral hygiene maintenance protocols were reinforced.

### 2.9. Power and Statistical Analyses

Sample-size estimation (nQuery Advisor 6.0, Statistical Solutions, Saugas, MA, USA) was performed using data from a pilot investigation. Power analysis was based on the supposition that a mean difference of 1 mm and 1 mm in PD and CAL should be detected for a standardized difference of 0.5 at a significance level of 0.01. It was estimated that with the inclusion of at least 18 CS, 18 ENDS-users, and 18 non-smokers, the study would achieve a power of 80%. Data were presented as means ± standard deviations and comparisons were conducted using the analysis of variance and Bonferroni *Post hoc* adjustment tests. The correlation between periodontal parameters and whole salivary cortisol and IL-1β levels was determined using logistic regression models. *p*-values that were <1% were selected as indicators of statistical significance.

## 3. Results

### 3.1. Demographics

Fifty-four individuals (18 CS, 18 ENDS users, and 18 non-smokers) were included. The mean age of CS, ENDS-users, and non-smokers was 45.6 ± 2.8, 41.3 ± 1.8, and 42.2 ± 3.5 years, respectively. CS had a smoking history of 12.7 ± 2.2 pack-years and ENDS-users were vaping for 6.8 ± 0.5 years. All non-smokers were former smokers and had cigarette-smoking 8.1 ± 0.5 years ago. Family history of smoking was reported by 11 (61.1%), 12 (66.7%), and 3 (16.7%) CS ENDS users and non-smokers, respectively. Most of the participants in all groups were males. Eleven (61.1%), 10 (55.6%), and 13 (72.2%) of CS, ENDS-users, and non-smokers stated that they were brushing their teeth twice a day. None of the participants had ever used dental floss (Table 1). None of the participants were aware of having psychological disorders (anxiety and/or depression). All ENDS users were using nicotine-containing e-liquids having an average nicotine concentration of 12.4 mg/mL.

### 3.2. Periodontal Parameters

The clinical AL, MT, PD, PI, and MBL were similar in all groups at baseline. Gingival bleeding was significantly high in non-smokers than CS (*p* < 0.01) and ENDS-users (*p* < 0.01) at baseline. At 12-weeks of follow-up, PI (*p* < 0.01) and PD (*p* < 0.01) were significantly high in CS and ENDS-users compared to non-smokers. There was no difference in GI and clinical AL in all patients at 12 weeks of follow-up (Table 2).

### 3.3. Whole Salivary Cortisol and IL-1β Levels

At baseline, whole salivary flow rate and cortisol and IL-1β levels were similar among CS, ENDS-users, and non-smokers. There was no statistically significant difference in whole salivary cortisol and IL-1β levels in CS and ENDS users at baseline and at 12-weeks’ follow-up. In non-smokers, there was a significant reduction in IL-1β (*p* < 0.01) and CL (*p* < 0.01) at 12 weeks of follow-up compared with baseline (Table 3).

### 3.4. Correlation between Whole Salivary Cortisol and IL-1β Levels

Among non-smokers, there was a statistically significant correlation between whole salivary cortisol and IL-1β levels at 12-weeks’ follow-up (*p* < 0.001) (Figure 1). There was no statistically significant correlation between whole salivary cortisol and IL-1β levels and gender, pack-years, duration and number of puffs among ENDS-users, age, and clinical periodontal parameters at baseline and 12-weeks’ follow-up.

## 4. Discussion

Tobacco smoking is a risk factor for periodontal diseases such as periodontitis [30,31]. It was therefore anticipated that periodontal inflammation would be worse and whole salivary cortisol and IL-1β levels would be high in CS and ENDS-users than in non-smokers; however, our results showed no difference in periodontal parameters and whole salivary cortisol and IL-1β levels in all groups at baseline. These outcomes should be carefully interpreted as some factors may have influenced the reported results. Individuals with a smoking history of 0.1 to 20, 20.1 to 40, and over 40 pack-years are classified as “light”, “moderate” and “heavy” smokers, respectively [32,33]. In the present investigation, all CS were “light-smokers” as they had a smoking history of approximately 12 pack-years. All ENDS users were former smokers who had quit cigarette smoking nearly 6 years ago and were vaping nearly 10 times daily with approximately four puffs per session. The average amount of nicotine present in conventional cigarettes varies among commercial brands [34]. According to Taghavi et al. [34], the amount of nicotine in cigarettes ranges between approximately 6 and 28 mg; however, all nicotine present in cigarette smoke is not usually inhaled. In the present investigation, commercial brands used by CS were not asked for; however, ENDS users were using e-liquids that had an average nicotine concentration of 12.4 mg/mL. Both CS and ENDS users had a relatively short history of smoking and vaping respectively. There is a possibility that the amount of nicotine present in each cigarette smoked by CS and during each session of vaping among ENDS users were similar. Moreover, although most of the participants stated that they brushed their teeth twice a day (before breakfast and going to sleep), other critical factors such as brushing time and technique play a role in achieving adequate plaque control [35]; which undesirably remained unexplored in the present investigation as it was beyond the scope of the present study. Furthermore, none of the participants had ever used dental floss and the contribution of this factor towards the initiation and progression of periodontal inflammation in the study population cannot be overlooked. These factors could have resulted in demonstrating similar periodontal parameters and CL among CS, ENDS users, and non-smokers at baseline.

In the present investigation, the radiographic examination was performed only at baseline. This was primarily performed to determine the extent of MBL in the patient population. Since the current investigation had a short-term follow-up (12-weeks), there was no ethical reason to expose patients to another round of radiation exposure. Nevertheless, with reference to the classification of periodontal and peri-implant diseases [36], our baseline radiographic evaluation showed that none of the patients had periodontitis. This was astounding as we expected baseline MBL to be high in at least CS compared with non-smokers based on previous reports [22,37]. As mentioned above, CS were mainly light smokers, and ENDS users also had a short vaping history. This seems to be a clarification for the comparable MBL in all groups. The same explanation can be proposed for the similarity in clinical AL in all groups throughout the study duration. There is a likelihood that MBL and clinical AL are high in heavy-smokers and whole salivary IL-1β and cortisol levels are high in heavy smokers (>40 pack-years) than in CS with a smoking history of up to 20 pack-years (light-smokers). This warrants additional studies. Nevertheless, the laboratory-based results of the current investigation showed that salivary immunoinflammatory response is worse in CS and ENDS users than non-smokers. This outcome supports previous studies [38,39] that have shown that vaping is by no means a safe replacement for smoking.

Assessment of cotinine levels in biological fluids such as blood, urine, and saliva are laboratory-based investigations that can verify the smoking status of self-reported tobacco smokers [40,41,42]. Here, it is also important to mention that levels of cotinine in serum, saliva, and urine samples obtained from traditional CS and ENDS-users have been shown to be similar [41,43]. In the current investigation, the authors relied on the questionnaire to determine the nicotine intake status of the study population. In other words, no laboratory-based investigations were performed to verify the non-smoking, smoking, or vaping status of the patient population. The main reason for this limitation is that verification of self-reported nicotine intake status via laboratory-based experiments was beyond the scope of the present study. Nevertheless, since clinical periodontal parameters were worse and whole salivary CL was significantly high in CS and ENDS users than non-smokers; it is anticipated CS and ENDS users had high yet similar concentrations of cotinine in saliva compared with non-smokers. Additional studies are needed to test this hypothesis. With regards to the study groups, a fourth group comprising of non-smokers with a healthy periodontal status was not included in the present study. A reason for this is that the focus of the present study was on cigarette-smoking and vaping habits in contrast to their corresponding controls, that is non-smokers. It is hypothesized that whole salivary cortisol and IL-1β levels would have been markedly low in non-smokers with a healthy periodontal status in case they were included and compared with non-smokers of the present patient population. Lastly, the potential relationship of gender on salivary CL could not be established in the present investigation despite the fact that there were twice as many female non-smokers compared with other groups. According to Liu et al. [44], salivary CL was markedly higher in males compared with females; which is evidently in contraindication to the present results. One clarification for this is that gender-wise, males dominated the patient population in each group compared with females. Further studies are needed to elucidate the impact of gender on the expression of cortisol in biological fluids including UWS.

## 5. Conclusions

In light CS and ENDS users without periodontal disease, clinical periodontal parameters and whole-salivary CL and Il-1β levels remain unchanged after NSPT.

## Figures and Tables

**Figure 1 ijerph-19-11290-f001:**
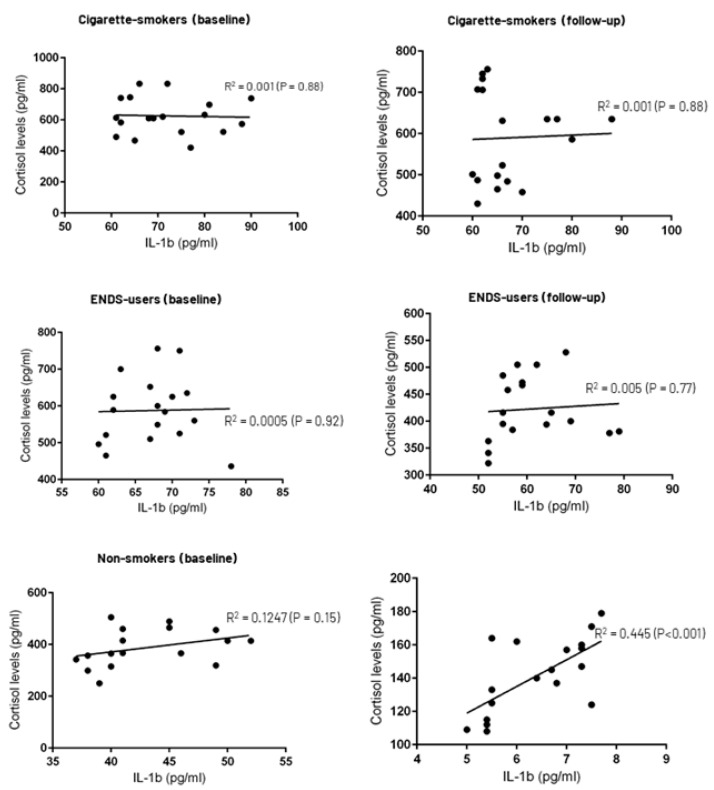
Correlation between whole salivary cortisol and IL-1β levels.

**Table 1 ijerph-19-11290-t001:** Demographics of the study cohort.

Parameters	Cigarette-Smokers	ENDS-Users	Non-Smokers
Patients (n)	18	18	18
Gender	14 males	12 males	10 males
	4 females	6 females	8 females
Mean age	45.6 ± 2.8 years	41.3 ± 1.8 years	42.2 ± 3.5 years
Duration of smoking pack years	12.7 ± 2.2 pack-years	NA	NA
Duration of vaping	NA	6.8 ± 0.5 years	NA
Puffs per vaping session	NA	4.3 ± 0.5 puffs/session	NA
Family history of smoking	11 (61.1%)	12 (66.7%)	3 (16.7%)
Tooth brushing			
Once daily	7 (38.9%)	8 (44.4%)	5 (27.8%)
Twice daily	11 (61.1%)	10 (55.6%)	13 (72.2%)
Flossing			
Once daily	None	None	None

NA: Not applicable.

**Table 2 ijerph-19-11290-t002:** Periodontal status at baseline and at 12 weeks of follow-up.

	Baseline			12-Weeks’ Follow-Up		
Parameters	Cigarette-Smokers (n = 18)	ENDS-Users (n = 18)	Non-Smokers (n = 18)	Cigarette-Smokers (n = 18)	ENDS-Users (n = 18)	Non-Smokers (n = 18)
Missing teeth (n)	5.2 ± 0.4 teeth	4.5 ± 0.1 teeth	4.4 ± 0.5 teeth	5.2 ± 0.4 teeth	4.5 ± 0.1 teeth	4.4 ± 0.5 teeth
Plaque index	2.1 ± 0.2	1.8 ± 0.2	2.1 ± 0.3	1.5 ± 0.3 ^†^	1.2 ± 0.04 ^†^	0.5 ± 0.007
Gingival index	0.5 ± 0.07 *	0.7 ± 0.05 *	2.5 ± 0.2	0.5 ± 0.04	0.5 ± 0.06	0.4 ± 0.003
Probing depth	4.5 ± 0.3 mm	4.4 ± 0.5 mm	4.6 ± 0.5 mm	2.5 ± 0.2 ^†^ mm	2.6 ± 0.3 ^†^ mm	0.6 ± 0.03 mm
Clinical attachment loss	1.7 ± 0.07 mm	1.4 ± 0.1 mm	1.4 ± 0.2 mm	1.7 ± 0.04 mm	1.5 ± 0.08 mm	1.4 ± 0.06 mm
Marginal bone loss (mesial surface)	3.2 ± 0.4 mm	2.9 ± 0.5 mm	2.7 ± 0.4 mm	NA	NA	NA
Marginal bone loss (distal surface)	3.3 ± 0.5 mm	3.04 ± 0.3 mm	2.7 ± 0.5 mm	NA	NA	NA

* Compared with non-smokers at baseline. ^†^ Compared with non-smokers at 12-weeks’ follow-up. NA: Not applicable. mm: millimeters.

**Table 3 ijerph-19-11290-t003:** Whole salivary flow rate and cortisol and interleukin 1-beta levels at baseline and at 12 weeks of follow-up.

	Baseline			12-Weeks’ Follow-Up		
Parameters	Cigarette-Smokers (n = 18)	ENDS-Users (n = 18)	Non-Smokers (n = 18)	Cigarette-Smokers (n = 18)	ENDS-Users (n = 18)	Non-Smokers (n = 18)
Salivary flow rate (mL/min)	0.15 ± 0.03 mL/min	0.13 ± 0.01 mL/min	0.11 ± 0.02 mL/min	0.13 ± 0.02 mL/min	0.12 ± 0.01 mL/min	0.12 ± 0.01 mL/min
Interleukin 1β (pg/mL)	72.2 ± 9.3 pg/mL	67.3 ± 5.1 pg/mL	42.8 ± 3.7 pg/mL *	54.1 ± 7.2 pg/mL	60.7 ± 5.5 pg/mL	6.4 ± 1.3 pg/mL
Cortisol levels (pg/mL)	625.4 ± 204.8 pg/mL	589.7 ± 153.3 pg/mL	386.4 ± 87.5 pg/mL	516.8 ± 143.8 pg/mL	422.8 ± 108.3 pg/mL	141.8 ± 33.7 pg/mL

* Compared with non-smokers at 12-weeks’ follow-up (*p* < 0.01). mm: millimeters.

## Data Availability

Data is available at reasonable request.

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
