# Peer review of "Comparison of Whole Salivary Cortisol and Interleukin 1-Beta Levels in Light Cigarette-Smokers and Users of Electronic Nicotine Delivery Systems before and after Non-Surgical Periodontal Therapy"

_ijerph, 2022, doi:10.3390/ijerph191811290_

Round 1
Reviewer 1 Report
Dear Authors, your study is well designed and the quality of the manuscript is high.
I suggest rewording the title to emphasise that the subjects were light smokers and had no periodontal disease. This aspect should also be highlighted in the conclusions.
My second comment concerns the gender distribution of the groups. There were twice as many women in the group of non-smokers as current smokers. This seems quite significant due to the effect of gender on cortisol secretion. This issue should be discussed.
I could not find Figure 1.
Author Response
Dear Authors, your study is well designed and the quality of the manuscript is high.
Response: Thank you for your time and critical feedback.
I suggest rewording the title to emphasise that the subjects were light smokers and had no periodontal disease. This aspect should also be highlighted in the conclusions.
Response: Thank you for the recommendation. We have revised the title and have emphasized that all cigarette-smokers were “light” smokers. As far as the periodontal status is concerned, we would like to clarify that we have no stringent criteria to include only patients without periodontal disease. This was a finding of the study. In other words, our results showed that all participants had a stable periodontal status. Nevertheless, we have revised the conclusion as follows:
In light CS and ENDS-users without periodontal disease, clinical periodontal parameters and whole-salivary CL and Il-1β levels remain unchanged after NSPT.
My second comment concerns the gender distribution of the groups. There were twice as many women in the group of non-smokers as current smokers. This seems quite significant due to the effect of gender on cortisol secretion. This issue should be discussed.
Response: Thank you for the comment. The potential effect of gender on cortisol levels has been discussed in the revised manuscript. An additional reference was added to the revised manuscript and the reference list has been adjusted accordingly.
I could not find Figure 1.
Response: Figure 1 has been added to the revised manuscript.
Reviewer 2 Report
Congratulations to the authors for the interesting work, we need more research regarding the use of electronic nicotine eletronic systems.
Author Response
Thank you for your time and valuable recommendations.
Reviewer 3 Report
The manuscript is written nicely, with few corrections and additions which will improve the quality of the paper and ease of understanding.
1. Paper is extensively based on statistical data but the data is not shown anywhere, it would be highly recommended if the authors add the statistical data and values in table format.
2. In lines 212-213 " At 12-weeks of follow-up, there was a significant reduction 212 in IL-1β (P<0.01) and CL (P<0.01) compared with baseline (Table 3)". Please clarify which authors are talking about which group.
3. Discussion Line 243-244, "the present results showed that periodontal parameters and 243 whole salivary cortisol and IL-1β levels were statistically comparable in all groups". SL and ILB levels are significantly different in CS/ENDS group and in the nonsmoker's group at 12 weeks of follow-up. Please clarify or explain in detail in this sentence.
4. I did not find figure 1 in the manuscript, was that uploaded separately?
Author Response
The manuscript is written nicely, with few corrections and additions which will improve the quality of the paper and ease of understanding.
Response: Thank you for your time and critical feedback.
- Paper is extensively based on statistical data but the data is not shown anywhere, it would be highly recommended if the authors add the statistical data and values in table format.
Response: Thank you for the comment The data related to correlation between cortisol and interleukin 1-beta has been added in the form of Figure 1. The related regression and p-values have also been entered. As far as the demographic data is concerned, it is shown in Table 1. Clinical data and related statistics with reference to periodontal status at baseline and at 12-weeks of follow-up is shown in Table 2. Furthermore, statistical comparisons related to whole salivary flow rate and cortisol and interleukin 1-beta levels at baseline and at 12-weeks of follow-up are shown in Table 3. The only data that is not shown is the correlation between whole salivary cortisol and IL-1β levels and gender, pack-years, duration and number of puffs among ENDS-users, age, and clinical periodontal parameters at baseline and 12-weeks’ follow-up. The reason for not showing this data is that there was no significant correlation and I added, then there would have been at least 5 tables or figures showed non-significant results. This, we clarified in the results that there was no statistically significant correlation between whole salivary cortisol and IL-1β levels and gender, pack-years, duration and number of puffs among ENDS-users, age, and clinical periodontal parameters at baseline and 12-weeks’ follow-up; and in parenthesis stated that this data is not shown.
- In lines 212-213 " At 12-weeks of follow-up, there was a significant reduction 212 in IL-1β (P<0.01) and CL (P<0.01) compared with baseline (Table 3)". Please clarify which authors are talking about which group.
Response: Thank you for pointing out this discrepancy. We have revised the sentence in the revised manuscript and have clarified that the text refers to non-smokers.
- Discussion Line 243-244, "the present results showed that periodontal parameters and 243 whole salivary cortisol and IL-1β levels were statistically comparable in all groups". SL and ILB levels are significantly different in CS/ENDS group and in the nonsmoker's group at 12 weeks of follow-up. Please clarify or explain in detail in this sentence.
Response: Thank you for the comment. The sentence has been clarified in the revised manuscript.
- I did not find figure 1 in the manuscript, was that uploaded separately?
Response: Figure 1 has been added to the revised manuscript.